# Simulation Research on Time-Optimal Path Planning of UAV Utilizing the Flightmare Platform

1st Yuling Xin
*School of Automation Engineering*
*University of Electronic Science*
*and Technology of China*
Chendu, China
xinyuling01@163.com

2nd Xin Lu
*Yangtze Delta Region Institute (Huzhou)*
*University of Electronic Science*
*and Technology of China*
Huzhou, China
luxin_uestc@163.com

3rd Fusheng Li*
*School of Automation Engineering*
*University of Electronic Science*
*and Technology of China*
Chendu, China
lifusheng@uestc.edu.cn

*Abstract*—This paper presents a study on time-optimal path planning and control for Unmanned Aerial Vehicles (UAVs) using fourth-order minimum snap trajectory generation and Nonlinear Model Predictive Control (NMPC) on the Flightmare simulation platform. Targeting the demands of fast flight in complex environments, a fourth-order polynomial trajectory planner is designed to minimize flight time while adhering to dynamical constraints. Integration with an NMPC and a PID controller enables precise tracking and dynamic adjustment of planned trajectories. Experimental results demonstrate that this method generates efficient and smooth flight trajectories, significantly reducing flight time while ensuring UAV stability and safety.

*Index Terms*—Flightmare Platform, Fourth-Order Minimum Snap Trajectory Generation, High-Fidelity Simulation, UAV, NMPC

## I. Introduction

As Unmanned Aerial Vehicle (UAV) technology continues to evolve at a rapid pace, its applications have broadened significantly across diverse fields. UAVs, also known as drones, have become indispensable tools for tasks requiring high-speed, agile, and autonomous responses [1]. These include but are not limited to package delivery, search-and-rescue operations, aerial photography, environmental monitoring, and even military applications [2]. Within these applications, the ability to plan time-optimal flight paths that align seamlessly with UAV dynamics is paramount for improving overall performance and safety.

Time-optimal path planning for UAVs is a complex problem that involves optimizing flight trajectories to minimize the total flight time while adhering to various constraints such as dynamical limitations, obstacle avoidance, and energy efficiency [3]. This optimization process not only ensures faster completion of missions but also enhances the stability and safety of the UAVs during operation.

Traditional approaches to path planning for UAVs often focus on generating collision-free paths, but they often fail to account for the intricate dynamics of the aircraft, leading to suboptimal flight performance [4]. To overcome this limitation, recent research has explored the integration of advanced trajectory planning and control techniques [9].

The fourth-order minimum snap trajectory generation method optimizes the snap term (fourth derivative of the

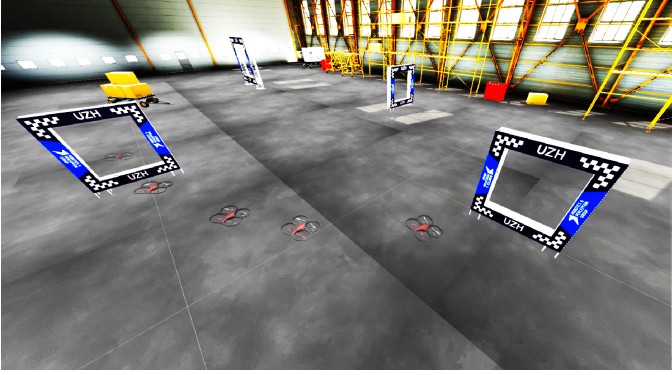

Fig. 1. Experimental results on the Flightmare simulation platform.

position) of the trajectory [15]. This approach ensures that the generated trajectories are both smooth and aggressive, which is crucial for achieving high-speed flight in complex environments. The integration of an NMPC and a PID controller further enhances the system's capabilities by dynamically adjusting control inputs based on real-time state feedback. This allows for precise tracking of the planned trajectory and resilience against uncertainties during flight.

The proposed framework is evaluated using the Flightmare simulation platform, a high-fidelity drone simulation based on the Unity engine. This platform offers precise physics modeling and flexible interfaces for algorithm development, making it an ideal testbed for validating the effectiveness of the proposed method. The experimental results demonstrate that the integration of fourth-order minimum snap trajectory generation with NMPC generates efficient and smooth flight trajectories, significantly reducing flight time while ensuring UAV stability and safety. The flightmre experimental results are shown in Figure 1.

## II. Problem Formulation

### A. Agile High-speed Flight

High-speed Unmanned Aerial Vehicles (UAVs) operating in complex environments face numerous challenges in trajectory generation and control. These challenges stem from the intricate dynamics of quadrotors, the stringent requirements on

agility, and the need to adapt quickly to unexpected obstacles and environmental changes [1].

In terms of trajectory generation, high-speed flight demands trajectories that are not only collision-free but also highly dynamic and aggressive to minimize flight time. Traditional methods of trajectory planning, such as spline interpolation or simple waypoint navigation, often fail to generate trajectories that fully exploit the full capabilities of the UAVs, particularly at high speeds [4]. Minimizing the flight time while adhering to strict dynamical constraints and avoiding obstacles becomes an NP-hard optimization problem that requires sophisticated algorithms to solve efficiently.

Control of high-speed UAVs further complicates the problem due to the inherent nonlinearities and uncertainties in the system dynamics. Real-time adjustments are crucial to handle external disturbances, actuator saturation, and sensor noise. Moreover, the fast-changing environment necessitates a control scheme that can rapidly replan and adjust the trajectory on the fly to ensure safety and mission success.

In summary, agile high-speed UAVs require:

1) Trajectory generation algorithms that can produce smooth yet aggressive trajectories to minimize flight time under strict dynamical and environmental constraints.

2) A robust control framework that can dynamically adjust control inputs based on real-time feedback to handle uncertainties and disturbances, ensuring precise tracking of the planned trajectory.

### B. Optimal Problem

Traditionally, optimal control problems in the context of UAVs aim to minimize a cost function subject to a set of constraints on the system dynamics and inputs. This formulation allows balancing multiple objectives, such as minimizing flight time, energy consumption, or control effort, while ensuring that the UAV operates within its physical and operational limits.

Mathematically, an optimal control problem can be formulated as follows:

$$\min_{\mathbf{u}} \quad \int_{t_0}^{t_f} \mathcal{L}_a\left(\mathbf{x}, \mathbf{u}\right) dt$$
$$\text{subject to} \quad \mathbf{r}(\mathbf{x}, \mathbf{u}, \mathbf{z}) = 0 \quad (1)$$
$$\mathbf{h}(\mathbf{x}, \mathbf{u}, \mathbf{z}) \leq 0$$

## III. DRONE MODELING

### A. Nomenclature

In this work, we establish a comprehensive mathematical framework for robot vision systems. We define a world frame $W$ with an orthonormal basis $\{x_W, y_W, z_W\}$ to represent the global environment. Additionally, a body frame $B$ with an orthonormal basis $\{x_B, y_B, z_B\}$ is introduced to describe the robot's orientation and position. The body frame is attached to the quadrotor, with its origin aligned with the center of mass as illustrated in Fig. 2.

Throughout the document, vectors are denoted in boldface with a prefix indicating the frame of reference and a suffix specifying the vector's origin and terminus. For example, $\mathbf{w}_{WB}$ represents the position vector of the body frame $B$ relative to the world frame $W$, expressed in the coordinates of the world frame.

To represent the orientation of rigid bodies, including the robot, we employ quaternions. The time derivative of a quaternion $\mathbf{q}_{WB} = (q_w, q_x, q_y, q_z)$ is governed by the skew-symmetric matrix $\Lambda(\omega)$, where $\boldsymbol{\omega}_B = (\omega_x, \omega_y, \omega_z)^T$ represents the angular velocity.

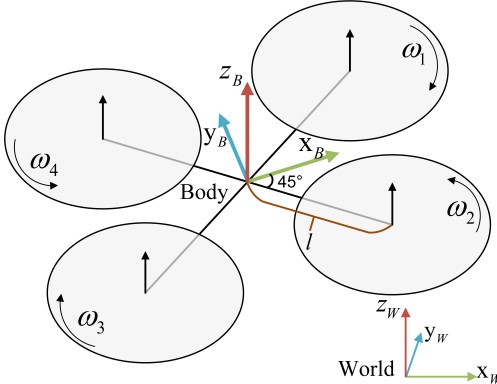

Fig. 2. Schematic diagrams of the quadrotor model being considered, along with the coordinate systems utilized.

### B. Quadrotor Dynamics

The drone is modeled as a rigid body with six degrees of freedom (DoF). The state vector $\mathbf{x} \in \mathbb{R}^{13}$ describing the evolution of the drone's configuration over time is given by:

$$\mathbf{x} = \begin{bmatrix} \mathbf{p}_{WB} \\ \mathbf{v}_{WB} \\ \mathbf{q}_{WB} \\ \boldsymbol{\omega}_B \end{bmatrix} \text{ and } \mathbf{u} = \begin{bmatrix} T \\ \boldsymbol{\tau} \end{bmatrix} \quad (2)$$

where: $\mathbf{p}_{WB} \in \mathbb{R}^3$ is the position of the drone's center of mass in the world frame $W$, $\mathbf{v}_{WB} \in \mathbb{R}^3$ is the linear velocity of the drone in the world frame, $\mathbf{q}_{WB} \in SO(3)$ is the quaternion representing the rotation from the body frame $B$ to the world frame $W$, $\boldsymbol{\omega}_B \in \mathbb{R}^3$ is the angular velocity of the drone in the body frame. $T$ is the total thrust produced by the drone's rotors, and $\boldsymbol{\tau}$ is the total torque acting on the drone.

$$\mathbf{J} = \begin{bmatrix} J_x & 0 & 0 \\ 0 & J_y & 0 \\ 0 & 0 & J_z \end{bmatrix} \quad (3)$$

where $J_x$, $J_y$, and $J_z$ are the moments of inertia of the drone about its principal axes.

$$T = \sum_{i=1}^{4} f_i \quad (4)$$

where $f_i$ is the thrust produced by the i-th rotor.

The time derivative of the state vector $\dot{\mathbf{x}}$ is governed by the following equations:

$$\dot{\mathbf{x}} = f(\mathbf{x}, \mathbf{u}) = \begin{bmatrix} \mathbf{v}_{WB} \\ \frac{1}{m}\left(m\mathbf{g}_W + \mathbf{q}_{WB} \odot \mathbf{T}_B\right) \\ \frac{1}{2}\mathbf{\Lambda}\left(\mathbf{\Omega_B}\right)\cdot\mathbf{q}_{WB} \\ \mathbf{J}^{-1}\left(\boldsymbol{\tau} - \boldsymbol{\omega}_B \times J\boldsymbol{\omega}_B\right) \end{bmatrix} \quad (5)$$

where: $\odot$ denotes the quaternion multiplication, $\mathbf{T}_B$ and $\boldsymbol{\tau}$ are the total force and torque acting on the drone, respectively, $m$ is the mass of the drone, $\mathbf{J} \in \mathbb{R}^{3\times 3}$ is the inertia matrix, $\mathbf{g}_W = [0, 0, -9.81]^T$ m/s² is the gravitational acceleration in the world frame.

The $\mathbf{\Lambda}$ means the skew-symmetric matrix of the angular velocity, which is given by:

$$\mathbf{\Lambda}\left(\omega\right) = \begin{bmatrix} 0 & -\omega_x & -\omega_y & -\omega_z \\ \omega_x & 0 & \omega_z & -\omega_y \\ \omega_y & -\omega_z & 0 & \omega_x \\ \omega_z & \omega_y & -\omega_x & 0 \end{bmatrix} \quad (6)$$

The torque $\boldsymbol{\tau}$ and total thrust $T$ are related to the individual i-th rotor thrust $f_i$ as:

$$\mathbf{T}_B = \begin{bmatrix} 0 \\ 0 \\ T \end{bmatrix} \text{ and } \boldsymbol{\tau} = \begin{bmatrix} \frac{l}{\sqrt{2}}(f_1 - f_2 - f_3 + f_4) \\ \frac{l}{\sqrt{2}}(-f_1 - f_2 + f_3 + f_4) \\ c_\tau(f_1 - f_2 + f_3 - f_4) \end{bmatrix} \quad (7)$$

## IV. Path Generation

In this section, we discuss the methods used for generating time-optimal paths for autonomous drone racing. Specifically, we focus on polynomial trajectory planning, particularly the use of fourth-order polynomials to minimize the snap of the trajectory, as this objective leads to aggressive and smooth trajectories suitable for drone racing.

### A. Polynomial Trajectory Planning

Polynomial trajectory planning leverages the differential flatness property of quadrotors to simplify full-state trajectory planning to a problem of planning only a few flat outputs (typically position and yaw) [14]. By representing the trajectory as a polynomial, we can efficiently compute the control inputs that achieve the desired trajectory [15].

*1) Minimizing Snap:* To generate aggressive and smooth trajectories, the objective is to minimize the snap (fourth-order derivative of position) of the trajectory [15] [16]. The snap $s(t)$ of a polynomial trajectory $p(t) = a_0 + a_1t + a_2t^2 + a_3t^3 + a_4t^4$ can be written as:

$$s(t) = p^{(4)}(t) = 24a_4t \quad (8)$$

where $p^{(4)}(t)$ denotes the fourth-order derivative of $p(t)$ with respect to time $t$.

The optimization problem can then be formulated as finding the polynomial coefficients $a_0, a_1, a_2, a_3, a_4$ that minimize the integral of the square of the snap over the trajectory duration $T$:

$$\min_{a_0,a_1,a_2,a_3,a_4} \int_0^T s(t)^2\, dt = \int_0^T (24a_4t)^2\, dt \quad (9)$$

However, in practice, we often minimize the maximum snap or add additional constraints and costs related to trajectory duration, smoothness, and feasibility. The full optimization problem includes constraints on the initial and final states of the drone (position, velocity, acceleration, and jerk) as well as any intermediate waypoints or obstacle avoidance constraints.

*2) Time Allocation:* Finding the optimal time allocation along the trajectory (i.e., determining how fast the drone should travel through each segment) is crucial for achieving minimum lap times. This is typically done by optimizing the polynomial coefficients jointly with the trajectory duration $T$:

$$\min_{a_0,a_1,a_2,a_3,a_4,T} \left( \int_0^T s(t)^2\, dt + \lambda \cdot T \right) \quad (10)$$

where $\lambda$ is a weight factor balancing the snap minimization and the total trajectory time.

### B. Implementation

Implementing a fourth-order polynomial trajectory planner involves solving the optimization problem described above. This can be done using numerical optimization techniques such as quadratic programming or nonlinear optimization solvers. The resulting trajectory is then used as a reference for the low-level controller to track.

In this paper, we adopt the polynomial trajectory planning approach to generate optimal paths. This method generates time-optimal trajectories by minimizing the snap of the trajectory.

In summary, polynomial trajectory planning with a focus on minimizing the snap of the trajectory is a powerful method for generating time-optimal and feasible paths for autonomous drone racing. This approach leverages the differential flatness property of quadrotors and enables the use of efficient optimization techniques to find optimal trajectories in real time.

## V. Model Predictive Control

Model Predictive Control (MPC) is a powerful technique for controlling complex systems with dynamical constraints [17]. For agile quadrotor flight, Nonlinear Model Predictive Control (NMPC) is particularly suited due to its ability to handle nonlinear dynamics and constraints effectively [9]. In this section, we detail the formulation and implementation of NMPC for quadrotor control.

### A. NMPC Formulation

The NMPC generates control inputs by solving a finite-time optimal control problem (OCP) over a receding horizon. The objective is to minimize the tracking error between the predicted states and reference states, while adhering to the system dynamics and constraints [5]. The optimization problem can be formulated as follows:

$$\mathcal{L}_a = \bar{\mathbf{x}}_N^T Q_N \bar{\mathbf{x}}_N + \sum_{i=1}^{N-1} \left( \bar{\mathbf{x}}_i^T Q_i \bar{\mathbf{x}}_i + \bar{\mathbf{u}}_i^T R_i \bar{\mathbf{u}}_i \right)$$

$$\text{s.t.}$$
$$\mathbf{x}_0 = \mathbf{x}_{\text{init}}, \qquad\qquad (11)$$
$$\mathbf{x}_{k+1} = f(\mathbf{x}_k, \mathbf{u}_k),$$
$$\mathbf{x}_k \in [\mathbf{x}_{\min}, \mathbf{x}_{\max}],$$
$$\mathbf{u}_k \in [\mathbf{u}_{\min}, \mathbf{u}_{\max}]$$

where $\bar{\mathbf{x}}_N^T Q_N \bar{\mathbf{x}}_N$ is the terminal cost, $\bar{\mathbf{x}}_i^T Q_i \bar{\mathbf{x}}_i$ and $\bar{\mathbf{u}}_i^T R_i \bar{\mathbf{u}}_i$ are the stage costs, $f(\mathbf{x}_k, \mathbf{u}_k)$ represents the discrete-time quadrotor dynamics, and $Q_i$, $R_i$, and $Q_N$ are positive definite weight matrices. The constraints ensure that the control inputs and angular velocities remain within specified bounds. And the $\bar{\mathbf{x}}$ and $\bar{\mathbf{u}}$ are defined as $\bar{\mathbf{x}} = \mathbf{x} - \mathbf{x}_{\text{ref}}$ and $\bar{\mathbf{u}} = \mathbf{u} - \mathbf{u}_{\text{ref}}$ respectively.

### B. Discretization of Dynamics

The continuous-time quadrotor dynamics need to be discretized for use in the NMPC framework. This can be achieved using numerical integration schemes such as Euler integration or Runge-Kutta methods. In our implementation, we use multiple-shooting as the transcription method and Runge-Kutta integration [18] to discretize the dynamics.

$$x_{k+1} = f_{\text{RK4}}(x_k, u_k, \Delta t) \qquad (12)$$

where $f_{\text{RK4}}$ is the Runge-Kutta 4th order integration function and $\Delta t$ is the discretization time step.

### C. Constraint Handling

Efficient constraint handling within the optimization framework is crucial for real-time performance. The NMPC formulation includes constraints on the angular velocities $\mathbf{\Omega}_{\mathbf{B}}$, thrust $T$, velocities $\mathbf{v}_{WB}$, and control inputs $\mathbf{u}$, ensuring that the control actions remain within the physical limits of the quadrotor.

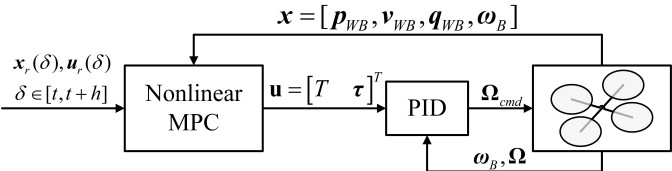

Fig. 3. Block diagram of the Nonlinear Model Predictive Controller with PID inner loop controller.

### D. Optimization Solver

The resulting nonlinear optimization problem is solved using a suitable solver, such as Sequential Quadratic Programming (SQP). In our implementation, we utilize the ACADO Toolkit [6] with qpOASES [7] as the underlying quadratic program solver.

### E. Integration with PID Controller

While NMPC provides a powerful framework for trajectory optimization and control, a PID controller can be used to complement the NMPC controller for enhanced stability and responsiveness. The PID controller can be used to regulate low-level system dynamics, such as the quadrotor's attitude, while the NMPC controller focuses on the high-level trajectory tracking. The integration of the two controllers is illustrated in Figure 3, where the NMPC controller generates the desired setpoints for the PID controller based on the time-optimal trajectory. The controller gains and parameters for the NMPC and PID controllers are summarized in Table I.

By integrating the PID and NMPC controllers, we can achieve a robust and responsive control system that can dynamically adjust to changes in the environment and mission requirements.

TABLE I
CONTROLLER GAINS AND PARAMETERS COMPARISON

| NMPC | | PID | |
|---|---|---|---|
| Parameter | Value | Parameter | Value |
| $Q$ | diag(200, 200, 500) | $K_p$ | 50 |
| $R$ | diag(10, 50) | $K_i$ | 1 |
| $dt$ | 50 ms | $K_d$ | 0.01 |
| N | 20 | | |

## VI. FLIGHTMARE

In this section, we introduce the Flightmare [8] simulation platform and discuss its advantages for validating the proposed time-optimal path planning and control framework. Flightmare is a high-fidelity quadrotor simulator designed for research and development, offering a range of features that make it an ideal testbed for evaluating UAV algorithms. We highlight the platform's unique capabilities and discuss the experimental setup used to validate the proposed method.

### A. Comparison of Quadrotor Simulators

In contrast to Hector [10], FlightGoggles [11], and AirSim [12] form Table II, Flightmare offers a unique combination of features that make it well-suited for UAV research. Flightmare's rendering engine is based on Unity, providing a flexible and high-speed rendering environment that can be tailored to the user's needs. The platform's physics simulation engine is highly configurable, supporting a range of dynamics from simple to real-world quadrotor behaviors. Flightmare is the only simulator among the compared ones that provides a point cloud extraction feature and an RL API, making it particularly suited for tasks requiring environmental 3D information and reinforcement learning-based control policies. Additionally, Flightmare can simulate multiple vehicles concurrently, facilitating research on multi-drone applications. All in all, Flightmare is chosen as the simulation platform for

| Simulator | Rendering | Dynamics | Sensor Suite | Point Cloud | RL API | Vehicles |
|---|---|---|---|---|---|---|
| Hector [10] | OpenGL | Gazebo-based | IMU, RGB | × | × | Single |
| FlightGoggles [11] | Unity | Flexible | IMU, RGB | × | × | Single |
| AirSim [12] | Unreal Engine | PhysX | IMU, RGB, Depth, Seg | × | × | Multiple |
| Flightmare [8] | Unity | Flexible | IMU, RGB, Depth, Seg | ✓ | ✓ | Multiple |

validating the proposed method due to its unique features and capabilities.

### B. Advantages of the Flightmare Platform

*1) Decoupled Rendering and Physics Engine:* One of the key strengths of Flightmare lies in its decoupled architecture, where the rendering engine based on Unity [19] is separated from the physics simulation engine. This design choice enables Flightmare to achieve remarkable performance: rendering speeds of up to 230Hz and physics simulation frequencies of up to 200,000Hz on a standard laptop [8]. This separation also allows users to flexibly adjust the balance between visual fidelity and simulation speed, tailored to the specific research needs.

*2) Flexible Sensor Suite:* Flightmare comes equipped with a rich and configurable sensor suite, including IMU, RGB cameras with ground-truth depth and semantic segmentation, range finders, and collision detection capabilities. This enables researchers to simulate a wide range of sensing modalities, critical for developing and testing perception-driven algorithms. Furthermore, Flightmare provides APIs to extract the full 3D point cloud of the simulated environment, facilitating path planning and obstacle avoidance tasks.

*3) Scalability and Parallel Simulation:* The platform's flexibility extends to supporting large-scale simulations, enabling the parallel simulation of hundreds of quadrotors. This feature is invaluable for reinforcement learning applications, where data efficiency is crucial. By simulating multiple agents in parallel, Flightmare allows for rapid data collection, significantly accelerating the training process for control policies.

*4) Open-Source and Modular Design:* Flightmare's open-source nature and modular design encourage collaboration and extendibility. The platform provides a clear and well-documented API, facilitating integration with existing research tools and libraries. The modular structure also makes it easy to swap out components, such as the physics engine or rendering backend, based on the specific research requirements. In this work, we use the RotorS [13] as the underlying quadrotor dynamics model in Flightmare, demonstrating the platform's flexibility and modularity.

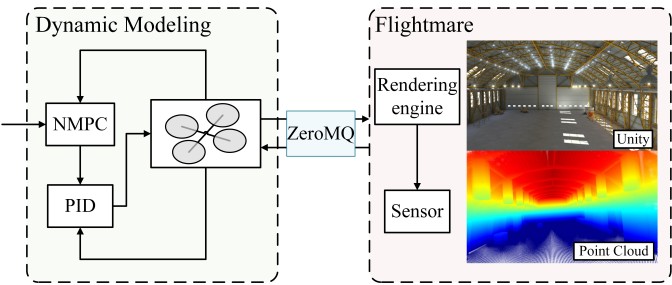

Fig. 4. Block diagram of the integration of control algorithms with Flightmare.

## VII. EXPERIMENTS

In this section, we present the experimental setup and results of the proposed time-optimal path planning and control framework for autonomous drone racing. The integration of polynomial trajectory planning and NMPC is . validated in a simulated environment using the Flightmare platform. The results demonstrate the effectiveness of the proposed method in generating efficient and smooth flight trajectories, enabling UAVs to navigate precisely and stably along planned paths.

### A. Experimental Setup

To evaluate the proposed time-optimal path planning and control framework in the flightmare simulation platform, we firstly design the control flow as shown in Fig. 4. The Flightmare decouples the rendering and physics engines, and the interface between the rendering engine and the quadrotor dynamics is implemented using the high-performance asynchronous messaging library ZeroMQ [20].

The quadrotor configurations used in the simulation are shown in Table III.

### B. Trajectory Tracking Performance on Giving Path

To evaluate the trajectory tracking performance of the proposed framework, we first consider a simple scenario where the drone is required to track a given path. The path is defined as a spiral ascent trajectory given by:

$$\mathbf{p}(t) = \begin{bmatrix} r(t)\cos(\omega t) \\ r(t)\sin(\omega t) \\ v_z t \end{bmatrix} \tag{13}$$

where $r(t) = r_0 + v_r t$ is the radius of the spiral, $\omega$ is the angular velocity, and $v_z$ is the vertical velocity. The drone

TABLE III
QUADROTOR CONFIGURATIONS

| Parameter(s) | Value(s) |
|:---:|:---:|
| $m$ [kg] | 0.6 |
| $l$ [m] | 0.125 |
| $J_x$ $[kg \cdot m^2]$ | 2.1e-3 |
| $J_y$ $[kg \cdot m^2]$ | 2.3e-3 |
| $J_z$ $[kg \cdot m^2]$ | 4.0e-3 |
| $(T_{\min}, T_{\max})$ [N] | (0, 8.5) |
| $c_\tau$ $[N \cdot m/(rad/s)^2]$ | 2.1e-6 |
| $c_T$ $[N/(rad/s)^2]$ | 1.2e-6 |

is required to track this path while maintaining a constant altitude.

The trajectory tracking performance of the proposed NMPC controller is shown in Fig. 5. In the figure, the pink dashed line represents the desired path, while the orange line represents the actual trajectory of the drone. The drone successfully tracks the spiral ascent trajectory, demonstrating the effectiveness of the proposed framework in generating smooth and accurate flight trajectories.

The error between the desired path and the actual trajectory is shown in Fig. 6. The error remains within an acceptable range, indicating that the drone is able to track the desired path accurately.

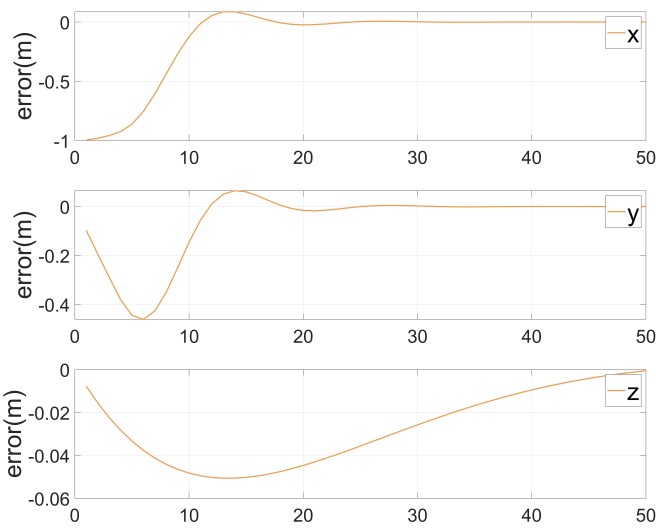

Fig. 6. Error between the desired path and the actual trajectory of the drone. The top, middle, and bottom plots represent the error in the $x$, $y$, and $z$ directions, respectively.

The time-optimal path planning results are shown in Fig. 7 and Fig. 8. In these figures, the orange dashed line represents the time-optimal path generated by the polynomial trajectory planner, which is shown in section IV. And the pink line represents the actual trajectory of the drone, which is controlled by the NMPC controller. The drone successfully navigates through the four gates in a time-optimal manner, demonstrating the effectiveness of the proposed framework in generating aggressive and smooth flight trajectories.

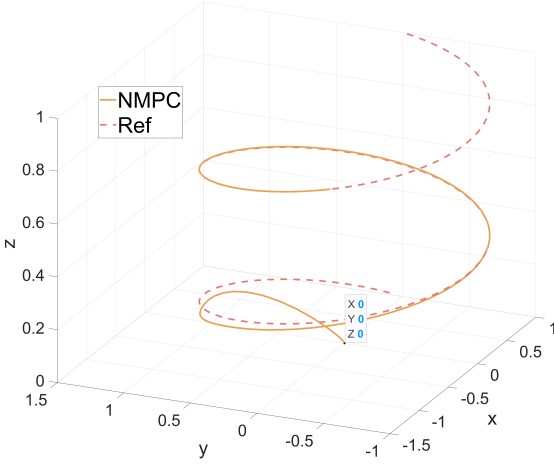

Fig. 5. Drone tracking the trajectory of a given spiral ascent path. The pink dashed line represents the desired path, while the orange line represents the actual trajectory of the drone.

### C. Time-Optimal Path Planning for NMPC Controller

In this experiment, the drone has to navigate through four gates in a time-optimal manner, which are placed at different locations in $(-10, 0, 2), (0, 10, 4), (10, 0, 2), (0, -10, 2)$ respectively.

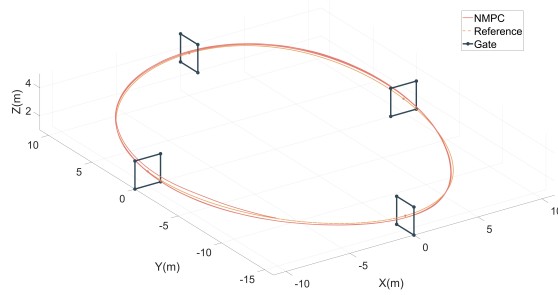

Fig. 7. Time-optimal path generation and NMPC tracking of the drone through four gates. The orange dashed line represents the time-optimal path, the pink line represents the actual tracking trajectory, and the four squares represent the positions of the gates.

The tracking performance from $x, y, z$ axis of the drone is shown in Fig. 9, which indicates that the drone can track the time-optimal path accurately from the $x, y, z$ axis.

### VIII. CONCLUSION

This paper presents a comprehensive framework for time-optimal path generation and control of Unmanned Aerial

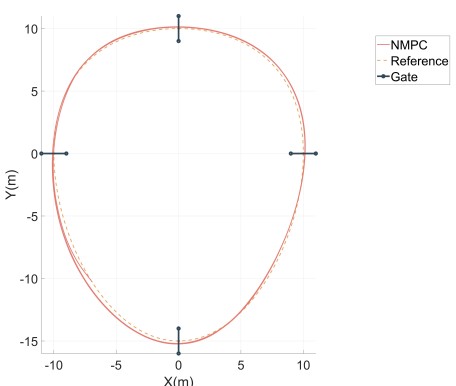

Fig. 8. Top view of the time-optimal path generation and NMPC tracking of the drone through four gates.

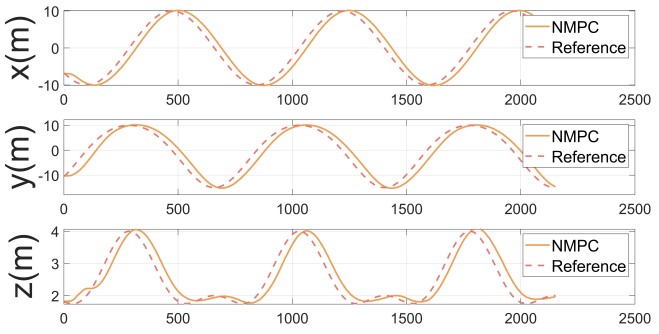

Fig. 9. Tracking performance of the drone through four gates in the $x, y, z$ axis. The top, middle, and bottom plots represent the tracking performance in the $x, y, z$ axis, respectively. The horizontal error indicates the control delay.

Vehicles (UAVs) using fourth-order minimum snap trajectory generation and Nonlinear Model Predictive Control (NMPC). The framework is designed to address the challenges of agile high-speed flight in auto race, aiming to minimize flight time while adhering to strict dynamical constraints.

The proposed method utilizes the fourth-order polynomial trajectory generation approach to generate smooth yet aggressive trajectories. By minimizing the snap term (fourth derivative of position), the generated trajectories are optimized for high-speed performance while ensuring their feasibility and safety. The integration of NMPC controller further enhances the system capabilities by dynamically adjusting control inputs based on real-time state feedback, enabling precise trajectory tracking and resilience against uncertainties during flight.

The effectiveness of the proposed framework is evaluated using the Flightmare simulation platform, a high-fidelity drone simulator based on the Unity engine. The experimental results demonstrate that the integration of fourth-order minimum snap trajectory generation with NMPC generates efficient and smooth flight trajectories, significantly reducing flight time while ensuring UAV stability and safety. This approach is well-suited for autonomous UAV operations in complex environments, such as drone racing and aerial photography.

Future work could further optimize the trajectory planning and control algorithms, explore adaptive control strategies, and investigate their application in real-world UAV platforms.

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
