# OpenReview forum: "Simulation Research on Time-Optimal Path Planning of  UAV Utilizing the Flightmare Platform"
_IEEE.org/ICIST/2024/Conference — IEEE ICIST 2024 Conference Submission_

### Official Review · Reviewer_pN47 · 2024-08-21
**Simulation Research on Time-Optimal Path Planning of UAV Utilizing the Flightmare Platform**

**Rating:** 7
**Confidence:** 4

**Review:**

This paper presents a study on time-optimal path planning and control for Unmanned Aerial Vehicles using fourth-order minimum snap trajectory generation and Nonlinear Model Predictive Control on the Flightmare simulation platform. The research results of this paper have great practical significance. However, the crux of the argument detailing the proposed algorithm could be presented better. Moreover, it would be better to include quantitative comparisons in the simulation results.

---

### Official Review · Reviewer_rpPF · 2024-08-24
**The paper is written clearly, exceptionally excellent.**

**Rating:** 8
**Confidence:** 3

**Review:**

This paper excels in terms of quality, clarity, originality, and significance, but I would still like to offer some suggestions.
1. Can the method proposed in this paper be applied to actual UAV systems?
2. What original contributions should the paper emphasize?

---

### Official Review · Reviewer_nXKM · 2024-08-25
**Explain what this paper is about  On the Flightmare simulation platform, the time-optimal path planning and control of UAVs were studied by using the fourth-order minimum bounce trajectory generation and nonlinear model predictive control (NMPC) technology. And the advantage of the paper usage method: while accurately ensuring the stability and safety of the UA V, the flight time is significantly reduced. At the same time, 1. Highlight the innovative work of the article. 2. Elaborate on the future work and the direction for improvement.**

**Rating:** 7
**Confidence:** 4

**Review:**

The work content and the advantages of the paper method: On the Flightmare simulation platform, the time-optimal path planning and control of the UAV are studied by using the fourth-order minimum bounce trajectory generation and nonlinear model predictive control (NMPC) technology. In order to meet the needs of fast flight in complex environments, a fourth-order polynomial trajectory planner was cleverly designed, which perfectly minimized the flight time under the premise of satisfying the dynamic constraints. Integration with NMPC and PID controllers allows for precise tracking and dynamic adjustment of planned trajectories. Experimental results show that the proposed method generates an efficient and smooth flight trajectory, which significantly shortens the flight time while accurately ensuring the stability and safety of UAV.
However：1. Highlight the innovative work of the article.
2. Elaborate on the future work and the direction for improvement.

---

### Decision · Program_Chairs · 2024-09-08

Accept (Oral)